# Omicron surge impact on acute kidney injury in ICU patients: A study using the ISARIC COVID-19 database

Danyang Dai[1]*, Pedro Franca Gois[2,3], Marina Wainstein[2], Moji Ghadimi[4],
Nicholas Spyrison[4], Rolando Claure-Del Granado[5,6], Sally Shrapnel[1,7,8], Jason D. Pole[1,8]*,
ISARIC Characterization Group[9]¶

**1** Centre for Health Services Research, University of Queensland, Brisbane, Australia, **2** Faculty of Health, Medicine & Behavioural Sciences, University of Queensland, Brisbane, Australia, **3** Nephrology and Transplantation, John Hunter Hospital, New South Wales, Australia, **4** School of Mathematics and Physics, University of Queensland, Brisbane, Australia, **5** Division of Nephrology Hospital Obrero No 2 – Caja Nacional de Salud, Cochabamba, Bolivia, **6** IIBISMED, Universidad Mayor de San Simon, School of Medicine, Cochabamba, Bolivia, **7** ARC Centre of Excellence for Engineered Quantum Systems, School of Mathematics and Physics, University of Queensland, Queensland, Australia, **8** Dalla Lana School of Public Health, The University of Toronto, Toronto, Ontario Canada, **9** International Severe Acute Respiratory and Emerging Infections Consortium (ISARIC), Pandemic Sciences Institute, University of Oxford, Oxford, United Kingdom

¶ The full list of ISARIC Characterization Group can be found in the acknowledgement.
* j.pole@utoronto.ca (JDP); danyang.dai@uq.edu.au (DD)

## Abstract

### Background

Acute Kidney Injury (AKI) is common among COVID-19 patients and is associated with a higher risk of death. Compared to earlier COVID-19 variants, Omicron has reduced mortality. To study the relationship between Omicron and AKI, we conducted the first international study using the global International Severe Acute Respiratory and Emerging Infection Consortium (ISARIC) COVID-19 global dataset.

### Methods

This prospective observational study aims to characterise AKI in a cohort of 3,908 COVID-19 patients admitted to the intensive care unit (ICU) across six countries. Clinical characteristics were compared between Omicron and pre-Omicron patients. Multivariable logistic regression was used to analyse the relationship between the Omicron variant and AKI. AKI was defined based on the change in serum creatinine levels, in accordance with the Kidney Disease Improving Global Outcome AKI guidelines.

### Results

Patients admitted to an ICU during the Omicron wave were older and had more comorbidities than pre-Omicron patients. The prevalence of AKI was the same

**Data availability statement:** The data that underpin this analysis are highly detailed clinical data on individuals hospitalised with COVID-19. Due to the sensitive nature of these data and the associated privacy concerns, they are available via a governed data access mechanism following review of a data access committee. Data can be requested via the IDDO COVID-19 Data Sharing Platform (http://www.iddo.org/covid-19).

**Funding:** UQ-QH Alliance Scholarship. SS is supported by ARC CoE Engineered Quantum Systems. DD, SS and NS declared funding from Artificial Intelligence for Pandemics (A14PAN) at University of Queensland. The ISARIC Clinical Characterisation Group is supported by the Wellcome Trust [303666/Z/23/Z]; UK International Development [301542-403]; and the Gates Foundation [INV-063472]. There was no additional external funding received for this study. The funders had no role in study design, data collection and analysis, decision to publish, or preparation of the manuscript.

**Competing interests:** Deputy Chair of the ISN Fellowship committee. This does not alter our adherence to PLOS ONE policies on sharing data and materials.

**Abbreviation:** AKI, Acute Kidney Injury; CA-AKI, Community-acquired AKI; CCP, Clinical Characterisation Protocol; CRF, Case Report Forms; eGFR, Estimated glomerular filtration rate; HA-AKI, Hospital-acquired AKI; ICU, Intensive Care Unit; IQR, Interquartile ranges; KRT, Kidney Replacement Therapy; SARS-Cov-2, Severe acute respiratory syndrome coronavirus 2; SMDs, Standardized mean differences; STROBE, Strengthening the Reporting of Observational Studies in Epidemiology; WHO, World Health Organization.

between Omicron and previous variants (24.7% vs 22.9%, p-value = 0.321). Controlling for confounders, ICU patients with the Omicron variant were 30%−40% less likely to develop AKI compared to patients with previous variants. The survival curves between AKI patients with Omicron and non-Omicron variants were consistent with the survival analysis.

## Conclusion

After adjusting for demographics, comorbidities, laboratory findings, and treatments, patients in ICU during the Omicron wave were less likely to develop AKI compared to previous eras. Nevertheless, the precise influence of the Omicron variant on kidney function remains a subject of ongoing discussion.

## Introduction

SARS-CoV-2 has, thus far, infected more than 769 million people and resulted in the loss of over 6.9 million lives [1]. In November 2021 Omicron (B1.1.529) was identified as a new variant of concern by the World Health Organization (WHO) [2]. Omicron was initially identified in South Africa and showed increased transmissibility [3], but reduced clinical severity and mortality compared to previous variants [3–7]. Acute kidney injury (AKI) has been recognized as one of the most prevalent and critical issues in patients admitted to hospital with COVID-19 [8–11] resulting in extended hospital stays, higher need for intensive care, and in-hospital death [8–11]. The prevalence of AKI among COVID-19 hospitalized patients between January 2020 and August 2020 was estimated to be 19.45% in a systematic review combining 40 studies [12].

Thus far, most of the COVID-19 and AKI studies have been limited to the earlier period of the pandemic, providing limited information about the relationship between Omicron and AKI development [12]. A single-centre study revealed a three-fold higher prevalence of AKI among intensive care unit (ICU) patients infected with the Omicron versus the Delta variant. Another study identified significant higher prevalence of AKI in the 4th wave which corresponds to the Omicron period compared to the 3rd wave [13]. In this study, the 4th wave AKI prevalence was lower compared to the first and second wave which corresponds to the beginning of COVID [13]. As both studies have limited patient cohorts, larger and more diverse population base is needed to further understand this relationship.

To further study the association between the Omicron variant and the prevalence of AKI in the ICU setting, we conducted the first multinational study focusing on AKI among ICU patients using the International Severe Acute Respiratory and Emerging Infections Consortium (ISARIC) COVID-19 database. We examined the prevalence of AKI and the characteristics and outcomes of patients admitted with various COVID-19 viral variants. Based on current literature, we hypothesized that patients infected with the Omicron variant would have a higher prevalence of AKI but a lower prevalence of in-hospital complications and death.

## Methods

### Study design

The findings in this paper utilized the ISARIC COVID-19 global database. This database includes clinically diagnosed or laboratory-confirmed SARS-CoV-2 infection data collected from 76 countries where 1,807 ISARIC partner institutions are located [14]. Since the beginning of the outbreak, the ISARIC-World Health Organisation (WHO) Clinical Characterisation Protocol (CCP) for Severe Emerging Infections has provided a framework for observational and prospective data collection on hospitalised patients using standardized case report forms (CRF) [15]. The protocol, CRF and study information can be found at https://isaric.org/research/covid-19-clinical-research-resources.[8,16,17]. Serum creatinine measurements were collected from all sites without standardizing time intervals, and each site had the freedom to determine the collection frequency [9].

In this prospective observational study, there were no changes made to clinical management practices. The protocol and consent forms can be accessed on the study website (https://isaric.net/ccp/). Given the overwhelming nature of the pandemic, while written consent was obtained in most sites, some sites approved oral consent or waiver of consent [8, 9]. Full details are available on the study website. The ISARIC-WHO CCP obtained approval from the World Health Organization Ethics Review Committee (RPC571 and RPC572, 25 April 2013). Ethical approval was obtained for each participating country and site following local guidance and requirements (Supplementary Statement in S1 File). This study follows the Strengthening the Reporting of Observational Studies in Epidemiology (STROBE) reporting guideline (S1 Table in S1 File).

### Study population

**Inclusion and exclusion criteria.** We included adult patients (aged 18 + years) from the ISARIC-WHO CCP database who were admitted to Intensive Care Units between June 1, 2021, and October 31, 2022, from countries reporting at least 10% of laboratory values for their patients. Criteria for clinical diagnosis can be found in S2 Table in S1 File. The study excluded non-ICU individuals; those who were identified as being on maintenance kidney replacement therapy (KRT) (dialysis or kidney transplantation); and patients who did not have a specified outcome (death, discharge or transfer). Patients with fewer than two serum creatinine measurements were also excluded from the study given that these were needed to determine AKI. To mitigate biases stemming from limited sample sizes, countries and regions (Brazil, Hong Kong, India, Kuwait, Libya and New Zealand) with fewer than 30 patients for both the pre-Omicron and Omicron periods were excluded.

Patients were categorized into two time periods: "Pre-Omicron" for admissions before December 1, 2021, and "Omicron" for admissions after January 1, 2022. These dates were chosen as they reflect the point at which the Omicron variant frequency reached 90% in the countries included in this study [4]. The exception to this was Pakistan which reported a peak Omicron frequency during mid-January 2022 [4]. December 2021 was considered a period of change in dominance of specific COVID-19 variants, where Omicron was spreading globally and incidence increasing while Delta was waning. As there were no clinical flags in the datasets that identified specific variants, it was not feasible to discriminate patients infected with the Omicron variant from pre-Omicron variants in December 2021 thus these data were excluded.

### AKI

AKI was defined according to the Kidney Disease Improving Global Outcome (KDIGO) guidelines serum creatinine criteria [18]. The KDIGO guidelines require a patient to have an increase in serum creatinine by 26.5 μmol/l within 48 hours or an increase to more than 1.5 times the baseline serum creatinine within 7 days. As most patients only have one reading of serum creatinine within 48 hours, an increase to more than 1.5 times the baseline serum creatinine within 7 days was used to identify AKI. As pre-admission serum creatinine values were unavailable, the first serum creatinine within 48 hours from admission was used as the baseline value [18]. KDIGO urine output criteria were not applied due to the infrequent collection of urine volume information in the CRF. The gradings of AKI were classified according to the KDIGO guidelines. AKI was categorised into community-acquired AKI (CA-AKI) if identified within the first 48 hours of admission and hospital-acquired AKI (HA-AKI) if it developed thereafter [19].

Patients' details, including comorbidities, medications, signs and symptoms, observations and laboratory test results were collected from the pre-specified CRF. Admission treatments, complications and outcomes were also reported. The outcomes included discharge, transfer to another hospital and in-hospital death. Comprehensive definitions for all recorded variables can be found in S3 Table in S1 File.

## Statistical Analysis

Descriptive statistical analysis was conducted with RStudio 2023.06.1+524 [20, 21]. Continuous variables were presented as medians and interquartile ranges (IQR), whereas categorical variables were depicted as counts and percentages. All statistical tests were performed using pairwise independent sample comparisons for the pre-Omicron cohort versus the Omicron cohort. The Mann-Whitney U test was performed for continuous variables and Pearson's Chi-squared test was used for categorical variables. Standardized mean differences (SMDs) were used to describe the differences between pre-Omicron patients and Omicron patients [22]. The relationship between in-hospital death and AKI is presented using a Kaplan-Meier survival curve and further stratified by viral variant [23]. The follow-up duration was calculated in days, starting from the first day of hospital admission and concluding with the earliest date of discharge, death or transfer to another hospital.

Logistic regression models were estimated to assess the relationship between AKI and the Omicron variant. Variables included in the logistic regression models are presented in Table 2 alongside odds ratios, p-values and confidence intervals for the Omicron variant. Sandwich estimator was adopted to adjust for the standard errors to account for potential correlation in covariates [24]. Variable selection was based on known predictors of AKI from the existing literature [25]. Variables with more than 20% missingness were excluded from the analysis. Missing values for other variables included were imputed using multiple imputations by chained equation [26]. A list of imputed variables can be found in S1 Fig in S1 File. We examined the relationship between AKI and the Omicron variant controlling for demographic variables, comorbidities, laboratory studies and treatments. Age was categorised into 10-year bins starting from 30 years of age, the reference group for the logistic regression is age from 18 to 29. The sex information pertains to the "biological sex assigned or determined at birth". The models were fitted to data from pre-Omicron and Omicron periods. The pre-Omicron period served as the reference group, allowing for a specific examination of the relationship between the Omicron variant and the risk of AKI.

## Ethics approval and consent to participate

Ethics Committee approval for this work was given by the World Health Organisation Ethics Review Committee (RPC571 and RPC572 on 25 April 2013). Institutional approval was additionally obtained by participating sites including the South Central Oxford C Research Ethics Committee in England (Ref 13/SC/0149) and the Scotland A Research Ethics Committee (Ref 20/SS/0028) for the United Kingdom and the Human Research Ethics Committee (Medical) at the University of the Witwatersrand in South Africa as part of a national surveillance programme (M160667) collectively representing the majority of the data. Other institutional and national approvals are in place as per local requirements.

## Results

Data were collected for 62,480 ICU-admitted patients from 47 countries from the 1st of June 2021 to the 31st of October 2022. A breakdown of exclusions can be found in Fig 1. After exclusions 3,908 patients were included in the analysis. Of these, 3,203 were from the pre-Omicron period and 705 were from the Omicron period. The final countries included in this study were Canada, United States (USA), United Kingdom (UK), Spain, Pakistan and Nepal. The distribution of included patients across the six countries is shown in Fig 2. The UK contributed the most patients during the pre-Omicron surge, and Pakistan contributed the highest counts of Omicron cases.

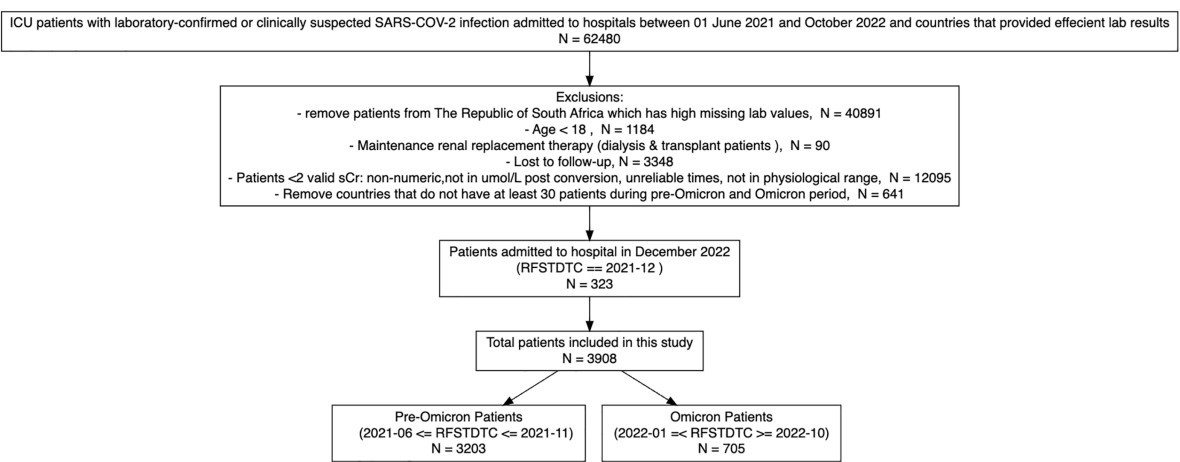

**Fig 1. Patients Inclusion and Exclusion Flowchart.** ICU: Intensive Care Unit; sCr – serum creatinine; RFSTDTC – Reference Start Date/Time, corresponds with the time and date of the subject's first study encounter for admission.

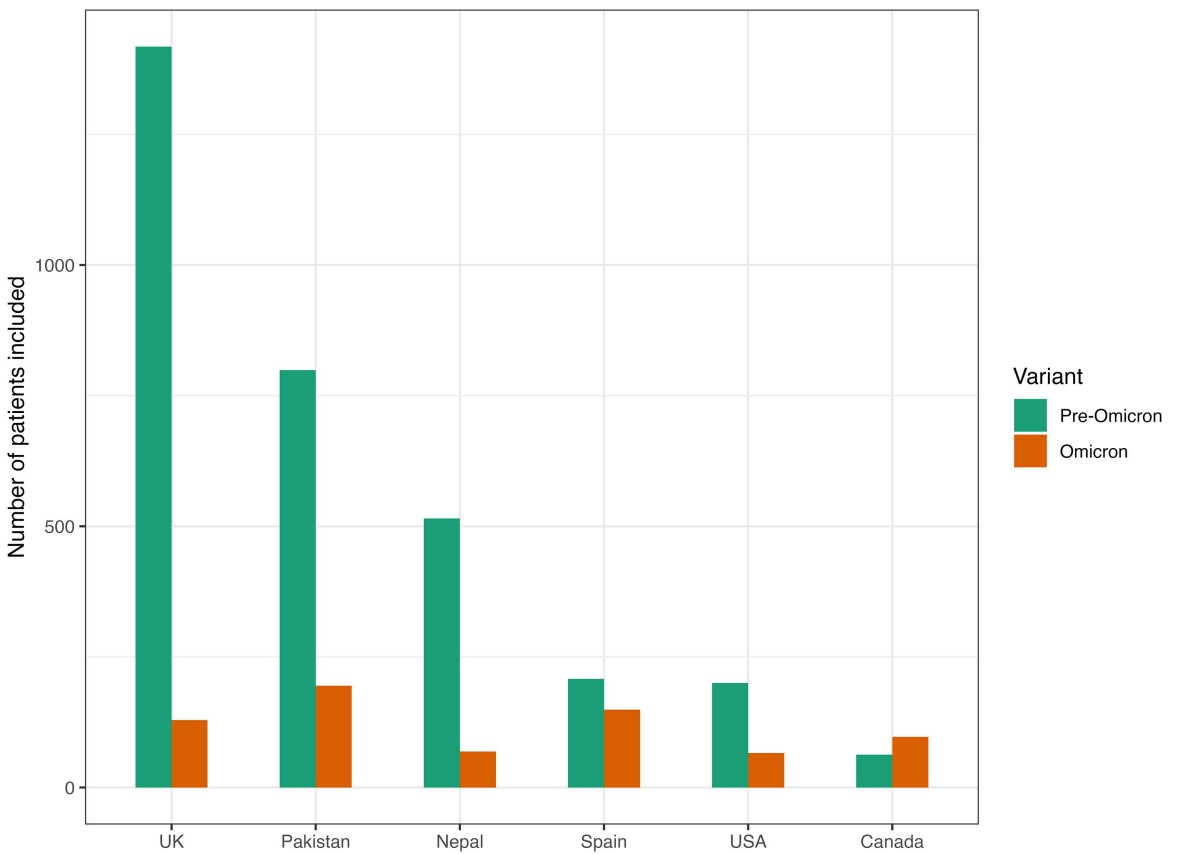

**Fig 2. ICU Patients Included in the Study by Countries and Variants.** ICU: Intensive Care Unit; USA: United States; UK: United Kingdom.

## Demographic and clinical characteristics

Baseline characteristics at hospital admission, acute interventions, complications and outcomes for pre-Omicron and Omicron ICU patients are shown in Table 1. The median age among patients admitted to ICU with COVID-19 pre-Omicron and Omicron variant was 57 (IQR: 45–68) and 63 (IQR: 52–74) years of age respectively. Less than half of the hospitalised patients were female (38% for pre-Omicron and 38% for Omicron). Omicron patients had a higher prevalence of comorbidities compared to pre-Omicron patients. The prevalence of chronic kidney disease was 12% for Omicron patients and 6% for pre-Omicron patients. The median serum creatinine on admission was 76 μmol/L (IQR: 58, 114) for pre-Omicron patients and 87 μmol/L (IQR: 63, 160) for Omicron patients. The median for initial estimated glomerular filtration rate (eGFR) was 88.7 ml/min/1.73m$^2$ (IQR: 52.7, 109.1) for pre-Omicron patients and 68.9 ml/min/1.73m$^2$ (IQR: 33.3, 97.4) for Omicron patients. Pulmonary symptoms like cough and shortness of breath were significantly less frequent in the Omicron variants than in the pre-Omicron variants (38% vs 64%, p-value < 0.001; 55% vs 75%, p-value < 0.001). Lower usage of invasive mechanical ventilation was observed among Omicron patients compared to pre-Omicron ICU patients (62% vs 72%, p-value < 0.001). Acute dialysis requirement among Omicron and pre-Omicron AKI patients was 45% and 52% respectively (p-value = 0.150).

## Outcomes

In-hospital mortality was 46% for pre-Omicron ICU patients and 35% for Omicron ICU patients. The median time for ICU hospital stay was 11 (IQR: 6–18) days for pre-Omicron patients and 10 (IQR: 5–19) days for Omicron patients. The Kaplan-Meier curve depicted in Fig 3 indicated that AKI patients exhibit lower survival rates in comparison to non-AKI patients (log-rank test: p-value < 0.001). Among AKI patients, there was no difference between Omicron patients and pre-Omicron patients (Fig 3). The log-rank tests indicated no statistical difference between Omicron and pre-Omicron AKI patients for survival curves (log-rank test: p-value = 0.200). For non-AKI patients, the Kaplan-Meier curve suggests that Omicron patients have a higher survival rate compared to pre-Omicron patients (log-rank test: p-value < 0.001).

## AKI and logistic regression

For pre-Omicron periods, 732 (22.85%) patients were observed to have developed AKI. During the Omicron surge, 174 (24.68%) out of 705 Omicron patients were observed to have developed AKI. There was no statistical difference in the prevalence of AKI between pre-Omicron and Omicron ICU patients (p-value = 0.321). There were no statistical differences for stage 1 and stage 2 AKI between pre-Omicron and Omicron ICU patients (p-value = 0.139 and p-value = 0.704, respectively). There was a higher prevalence of stage 3 AKI patients observed in Omicron surge compared to pre-Omicron (p-value = 0.006). Fig 4 shows the AKI positive percentage breakdown in different countries. The lower-income countries (Pakistan and Nepal) had consistently higher AKI prevalence for pre-Omicron and Omicron periods compared to other countries (Fig 4). The USA and Canada had a lower AKI prevalence during the Omicron period (Fig 4). Most ICU patients with AKI were classified as AKI grade 3, accounting for 73.5% during the pre-Omicron period and 73.0% during the Omicron period (Fig 5). There was a consistent distribution of AKI grades among ICU patients between the pre-Omicron and Omicron periods (Fig 5). The chi-squared test between AKI grades and variant has shown no difference between the AKI grades between the pre-Omicron variant and Omicron variant (p-value = 0.635). Additionally, more than half of AKI patients experienced hospital-acquired AKI (HA-AKI) regardless of the viral variant, as shown in Fig 6. Table 2 presents the findings from multivariable logistic models addressing the association between Omicron and AKI. Multiple models were fit to include different control variables with AKI positive as the outcome in each model. Using the imputed data, Model 1 and Model 2 suggest that controlling for demographics variables and initial serum creatinine, Omicron patients admitted to ICU were more than 30% less likely to develop AKI. While after additionally controlling for comorbidities, lab results and treatments, there was no significant effect on whether the ICU patients had the Omicron or pre-Omicron variant on AKI.

**Table 1. Included Patients Characteristics.**

| | Pre-Omicron | | | | Omicron | | | | Test Statistics between Pre-Omicron and Omicron All Patients | |
|---|---|---|---|---|---|---|---|---|---|---|
| | All (2+sCr) | AKI | No AKI | Missing (%) | All (2+sCr) | AKI | No AKI | Missing (%) | Significant | SMDs |
| | 3203 | 732 | 2471 | | 705 | 174 | 531 | | | |
| **Demographics** | | | | | | | | | | |
| Age, year, median (IQR) | 57 (45, 68) | 59 (48, 69) | 57 (44, 67) | 0 | 63 (52, 74) | 62 (50.2, 73) | 64 (53, 74) | 0 | p<0.001 | 0.382 |
| Female (%) | 1,231 (38) | 261 (36) | 970 (39) | 0.09 | 268 (38) | 57 (33) | 211 (40) | 0.28 | p<0.001 | −0.007 |
| **AKI Grades & RRT, n (%)** | | | | | | | | | | |
| AKI grade 1 | 105 (3) | 105 (14) | 0 (0) | 0 | 29 (4) | 29 (17) | 0 (0) | 0 | 0.139 | 0.044 |
| AKI grade 2 | 89 (3) | 89 (12) | 0 (0) | 0 | 18 (3) | 18 (10) | 0 (0) | 0 | 0.704 | 0.014 |
| AKI grade 3 | 159 (5) | 159 (22) | 0 (0) | 0 | 48 (7) | 48 (28) | 0 (0) | 0 | 0.006 | 0.078 |
| AKI grade 3 – RRT | 379 (12) | 379 (52) | 0 (0) | 0 | 79 (11) | 79 (45) | 0 (0) | 0 | 0.565 | 0.019 |
| **Comorbidities**, n (%)** | | | | | | | | | | |
| Chronic Kidney Disease | 189 (6) | 81 (11) | 108 (4) | 12.52 | 83 (12) | 29 (17) | 54 (10) | 3.83 | p<0.001 | 0.188 |
| Chronic Cardiac Disease | 305 (10) | 86 (12) | 219 (9) | 12.43 | 106 (15) | 19 (11) | 87 (16) | 3.83 | p<0.001 | 0.141 |
| Chronic Pulmonary Disease | 211 (7) | 47 (6) | 164 (7) | 12.21 | 63 (9) | 6 (3) | 57 (11) | 4.26 | 0.029 | 0.066 |
| Hypertension | 1,186 (37) | 334 (46) | 852 (34) | 12.11 | 321 (46) | 94 (54) | 227 (43) | 3.4 | p<0.001 | 0.101 |
| Diabetes – Type 1&2 | 452 (14) | 84 (11) | 368 (15) | 53.79 | 130 (18) | 20 (11) | 110 (21) | 41.56 | 0.06 | 0.057 |
| Liver Disease | 55 (2) | 19 (3) | 36 (1) | 12.55 | 44 (6) | 7 (4) | 37 (7) | 3.97 | p<0.001 | 0.227 |
| Obesity | 505 (16) | 92 (13) | 413 (17) | 16.39 | 94 (13) | 19 (11) | 75 (14) | 8.79 | p<0.001 | 0.114 |
| **Signs & Symptoms on Admission, n (%)** | | | | | | | | | | |
| Altered consciousness/confusion | 189 (6) | 49 (7) | 140 (6) | 13.49 | 67 (10) | 11 (6) | 56 (11) | 27.66 | p<0.001 | 0.212 |
| Diarrhoea | 448 (14) | 83 (11) | 365 (15) | 12.33 | 46 (7) | 16 (9) | 30 (6) | 27.94 | p<0.001 | 0.210 |
| Fever | 2,074 (65) | 477 (65) | 1,597 (65) | 10.52 | 269 (38) | 78 (45) | 191 (36) | 26.95 | p<0.001 | 0.425 |
| Vomiting/nausea | 418 (13) | 68 (9) | 350 (14) | 12.46 | 52 (7) | 12 (7) | 40 (8) | 27.66 | p<0.001 | 0.143 |
| Muscle aches/joint pain | 480 (15) | 87 (12) | 393 (16) | 15.99 | 60 (9) | 11 (6) | 49 (9) | 29.5 | p<0.001 | 0.162 |
| Headache | 266 (8) | 33 (5) | 233 (9) | 15.8 | 37 (5) | 4 (2) | 33 (6) | 29.5 | 0.020 | 0.086 |
| Cough | 2,045 (64) | 427 (58) | 1,618 (65) | 9.71 | 267 (38) | 65 (37) | 202 (38) | 27.23 | p<0.001 | 0.391 |
| Shortness of breath | 2,388 (75) | 540 (74) | 1,848 (75) | 8.71 | 391 (55) | 108 (62) | 283 (53) | 25.53 | p<0.001 | 0.175 |
| **Observations on Admission, median (IQR)** | | | | | | | | | | |
| Temperature, C | 37.2 (36.6, 38) | 36.9 (36.4, 37.6) | 37.3 (36.7, 38.1) | 32.63 | 36.7 (36.3, 37.2) | 36.6 (36.2, 37.2) | 36.8 (36.3, 37.2) | 39.15 | p<0.001 | −0.526 |
| Systolic BP, mmHg | 127 (113, 140) | 129 (114, 141) | 126 (113, 140) | 3.4 | 128 (110, 144) | 130 (110, 145) | 127 (110, 144) | 8.51 | p<0.001 | 0.041 |
| Diastolic BP, mmHg | 75 (66, 83) | 75 (65, 86) | 75 (67, 82) | 3.9 | 72 (60, 83) | 70 (60, 84) | 73 (61, 82.5) | 9.36 | p<0.001 | −0.158 |
| Heart rate, BPM | 93 (82, 109) | 96 (82.5, 110) | 93 (82, 108) | 3.18 | 94 (79, 109) | 95 (85, 109.2) | 92 (78, 109) | 6.81 | p<0.001 | −0.047 |
| Respiratory rate, per min | 24 (20, 30) | 26 (21, 31) | 24 (20, 30) | 6.84 | 22 (20, 28) | 24 (20, 29) | 22 (20, 28) | 20.28 | p<0.001 | −0.267 |

*(Continued)*

| | Pre-Omicron | | | | Omicron | | | | Test Statistics between Pre-Omicron and Omicron All Patients | |
|---|---|---|---|---|---|---|---|---|---|---|
| | All (2+sCr) | AKI | No AKI | Missing (%) | All (2+sCr) | AKI | No AKI | Missing (%) | Significant | SMDs |
| | 3203 | 732 | 2471 | | 705 | 174 | 531 | | | |
| Oxygen saturation, % | 93 (89, 96) | 92 (88, 96) | 93 (90, 96) | 7.68 | 95 (91, 97) | 95 (91, 96) | 95 (91, 97) | 11.21 | p<0.001 | 0.268 |
| **Laboratory Results on Admission, median (IQR)** | | | | | | | | | | |
| Potassium (mmol/L) | 4.1 (3.7, 4.5) | 4.2 (3.8, 4.7) | 4.1 (3.7, 4.4) | 2.31 | 4.2 (3.8, 4.6) | 4.2 (3.8, 5) | 4.2 (3.8, 4.6) | 0.71 | p<0.001 | 0.197 |
| Serum Creatinine (umol/L) | 76 (58, 114) | 196.5 (99, 367.2) | 69 (54.8, 89) | 0 | 87 (63, 160) | 267 (133.1, 362.4) | 76 (60, 110.5) | 0 | p<0.001 | 0.129 |
| Serum Creatinine (umol/L) – CKD | 151 (102, 317) | 363 (181.2, 548) | 111.7 (88, 151) | 0 | 163 (108.9, 269.6) | 318.2 (194.5, 443) | 134.8 (97.5, 173.5) | 0 | 0.931 | −0.145 |
| eGFR (ml/min/1.73m2) | 88.7 (52.7, 109.1) | 27 (13.1, 63.7) | 95.8 (72.7, 112.3) | 0 | 68.9 (33.3, 97.4) | 21.2 (13.6, 47.5) | 82 (52, 102.8) | 0 | p<0.001 | −0.328 |
| Hemoglobin (g/L) | 133 (118, 146) | 124 (106, 140) | 135 (121, 148) | 5.15 | 123 (105, 139) | 113 (96, 132) | 127 (109.5, 143) | 20 | p<0.001 | −0.389 |
| Sodium (mmol/L) | 136 (133, 140) | 137 (134, 141) | 136 (133, 139) | 2.78 | 137 (134, 140) | 138 (135, 142) | 137 (134, 140) | 1.28 | p<0.001 | 0.107 |
| Platelets (10^9/L) | 200 (153, 261.8) | 199 (149, 269.5) | 200 (154, 259) | 12.89 | 220 (168, 291) | 205 (164.5, 279.5) | 227.5 (168, 292.8) | 15.32 | p<0.001 | 0.188 |
| WBC (10^9/L) | 8.1 (5.6, 11.9) | 10.9 (6.8, 15.6) | 7.7 (5.5, 11) | 18.51 | 9.6 (6.5, 14.2) | 10.2 (7, 15.1) | 9.4 (6.3, 13.6) | 10.64 | p<0.001 | 0.251 |
| **In hospital Treatment, n (%)** | | | | | | | | | | |
| Antiviral and COVID-19 targeting agents | 1,568 (49) | 357 (49) | 1,211 (49) | 0.78 | 288 (41) | 85 (49) | 203 (38) | 1.13 | p<0.001 | 0.162 |
| Antibiotic agents | 2,865 (89) | 707 (97) | 2,158 (87) | 0.56 | 606 (86) | 164 (94) | 442 (83) | 0.99 | p<0.001 | 0.098 |
| Antifungal agents | 332 (10) | 79 (11) | 253 (10) | 3.28 | 77 (11) | 18 (10) | 59 (11) | 1.13 | 0.755 | 0.011 |
| Corticosteroids | 2,836 (89) | 620 (85) | 2,216 (90) | 0.28 | 517 (73) | 146 (84) | 371 (70) | 0.85 | p<0.001 | 0.388 |
| Invasive mechanical ventilation | 2,311 (72) | 668 (91) | 1,643 (66) | 0.31 | 434 (62) | 138 (79) | 296 (56) | 0.14 | p<0.001 | 0.230 |
| vasopressors | 1,162 (36) | 504 (69) | 658 (27) | 2.59 | 252 (36) | 93 (53) | 159 (30) | 1.56 | 0.535 | 0.019 |
| **Complications**, n (%)** | | | | | | | | | | |
| Bacterial pneumonia | 413 (13) | 88 (12) | 325 (13) | 12.39 | 80 (11) | 13 (7) | 67 (13) | 7.8 | 0.028 | 0.071 |
| Cardiac arrest | 209 (7) | 91 (12) | 118 (5) | 8.68 | 66 (9) | 25 (14) | 41 (8) | 4.11 | 0.001 | 0.094 |
| Coagulation disorder | 123 (4) | 26 (4) | 97 (4) | 11.36 | 21 (3) | 6 (3) | 15 (3) | 7.8 | 0.085 | 0.058 |
| Rhabdomyolysis | 10 (0) | 6 (1) | 4 (0) | 11.49 | 2 (0) | 0 (0) | 2 (0) | 8.51 | 1.000 | 0.007 |
| **Outcomes, n (%)** | | | | | | | | | | |
| Transferred | 216 (7) | 30 (4) | 186 (8) | 0 | 51 (7) | 12 (7) | 39 (7) | 0 | 0.547 | 0.019 |
| Discharged | 1,501 (47) | 117 (16) | 1,384 (56) | 0 | 410 (58) | 55 (32) | 355 (67) | 0 | p<0.001 | 0.228 |
| Death | 1,486 (46) | 585 (80) | 901 (36) | 0 | 244 (35) | 107 (61) | 137 (26) | 0 | p<0.001 | 0.242 |
| Length of Stay (median, IQR) | 11 (6, 18) | 9 (5, 17) | 11 (7, 19) | 0.91 | 10 (5, 19) | 7 (4, 13.2) | 11 (5, 21) | 0.71 | p<0.001 | 0.017 |

eGFR: estimated glomerular filtration rate, sCr: serum creatinine, AKI: Acute Kidney Injury, SMD: standardized mean difference, WBC: white blood count

**: Comprehensive definitions for comorbidities, complication and outcomes can be found in S3 Table in S1 File

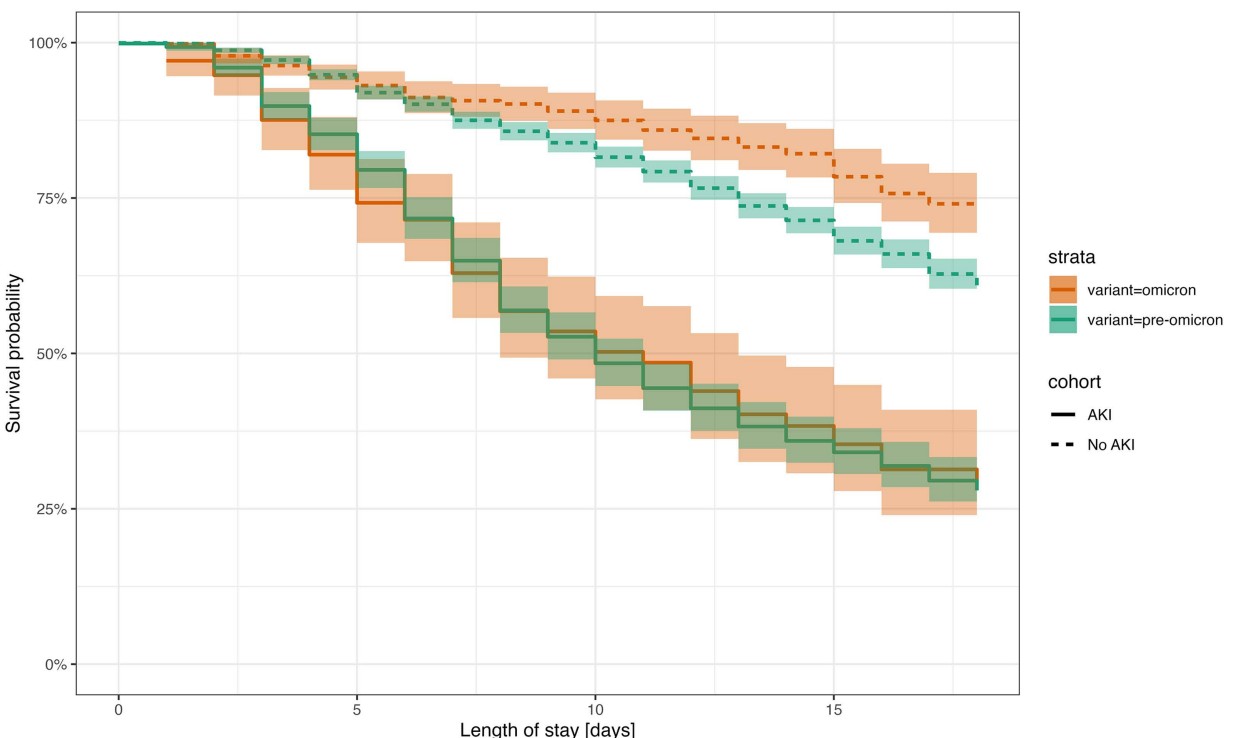

**Fig 3. Kaplan–Meier Survival plot stratified across acute kidney injury and variant.** Confidence bars are used to illustrate a 95% confidence interval. AKI, acute kidney injury.

Sensitivity analysis was carried out to examine the regressions. Using the listwise deletion method [27], model 1 and model 2 indicate consistent estimation compared to imputed values. Model 3, however, suggests a 60% decrease in the likelihood of developing AKI for the Omicron variant compared to the pre-Omicron variant. Looking into the imputed values which include lab values, all were within the correct range. When examining the occurrences of the missing values, Nepal had more than 99% missing in white blood cell count (WBC) and Spain had 78% missing values in haemoglobin (HGB) (S-Table 2). To adjust for the noise introduced by imputing large amounts of WBC and HGB values for Nepal and Spain, models were fit using the imputed data excluding Nepal and Spain. Using imputed data without Nepal and Spain, model 3 suggests that controlling for initial serum creatinine, demographics, comorbidities, lab results and treatments, patients admitted to the critical care unit with Omicron variant were more than 40% less likely to develop AKI compared to pre-Omicron variant.

## Discussion

This study is the first global investigation of the relationship between the Omicron variant and the characteristics and outcomes of hospitalized individuals with AKI and COVID-19 in the ICU setting. Since most COVID-19 AKI studies had focused on the onset of the COVID-19 pandemic, the prevalence of complications, such as AKI, associated with the Omicron variant has not been extensively studied. Our study encompassed a significantly diverse ICU population utilizing the ISARIC global COVID-19 dataset, with a total of 3,908 observations across six countries, allowing us to analyse the relationship between Omicron and AKI from a multinational standpoint. This multicentre global study has an overall similar prevalence of AKI among patients between the Omicron variant and Pre-Omicron variant. We observed a lower mortality, a lower prevalence of pulmonary embolism and a higher number of comorbidities among the Omicron variant in relation

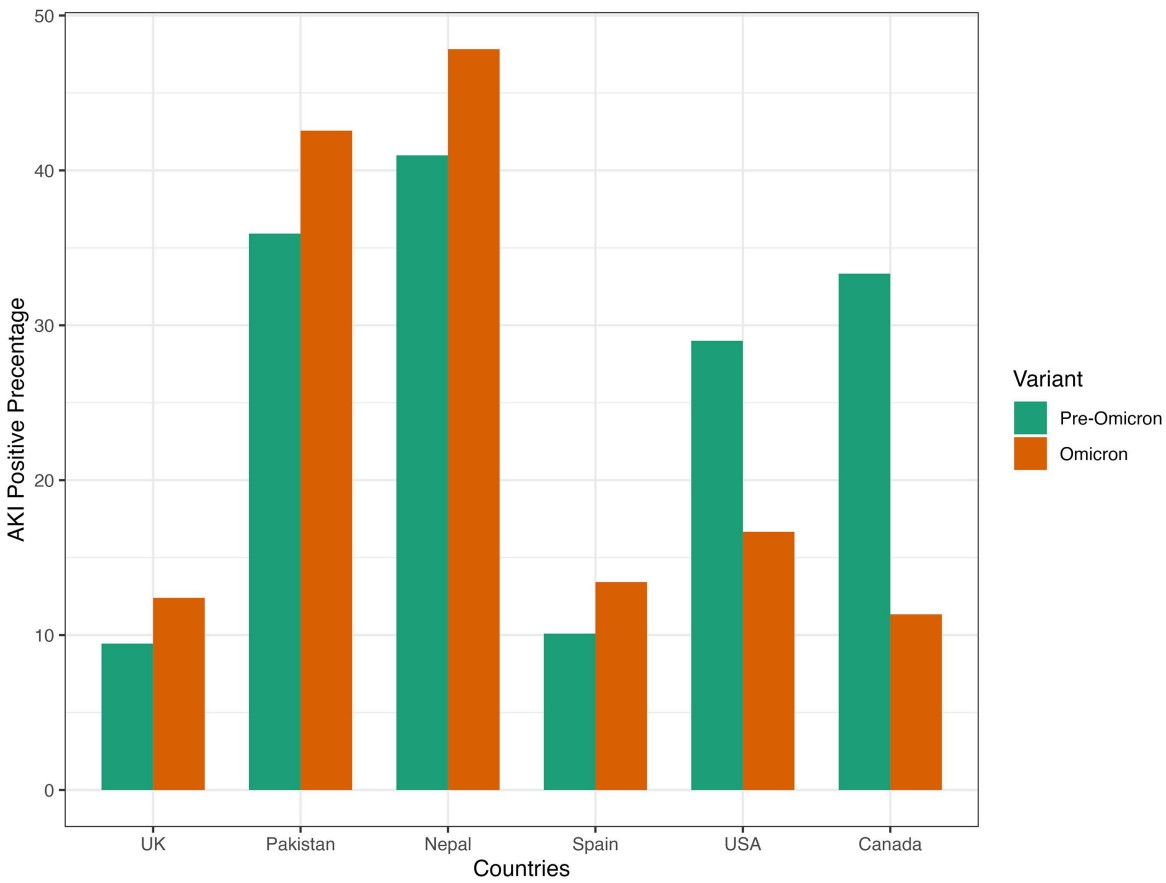

**Fig 4. AKI Positive Percentage by Countries and Variants.** AKI: acute kidney injury; USA: United States; UK: United Kingdom.

to previous variants for both AKI and no AKI patients. The logistic regression for AKI using the listwise deletion method and imputed data results in different odds ratio for Omicron and pre-Omicron variants. However, all models suggest a decrease in the likelihood of developing AKI for Omicron patients when controlling for initial serum creatinine, demographics, comorbidities, lab results and treatments.

Omicron infection has been established as having milder clinical characteristics and rapid transmission rates [4, 28]. Omicron variants have also been shown to have a lower risk of hospitalisation and reduced severity [29, 30]. Moreover, there has been a reduction in ICU admission during the Omicron wave compared to previous variants [30, 31]. The current understanding of AKI prevalence in Omicron patients is based on findings from two single-centre studies conducted in Italy and China. The Italian study reported 38.24% of ICU patients with Omicron developed AKI [32]. The Chinese study reported one AKI patient for Omicron hospitalised patients and no AKI patient was recorded for Delta patients [33]. Within our own cohort of ICU patients, we observed that the prevalence of AKI among those with Omicron was 24.68%. Single-centre studies can be restricted in their conclusion due to limited observation and unique regions. Some studies have reported that Omicron hospitalised patients were younger compared to previous variants [29, 33, 34], while others found patients were older [4, 32]. This is consistent with our study where Omicron patients were older. While many studies have established that Omicron is a milder version of COVID-19, the data were often limited to assessing the development of AKI in the context of emerging variants. There have been many studies have explored the risk factors associated with AKI development and COVID-19. One study in the USA has previously developed and validated a model to predict AKI

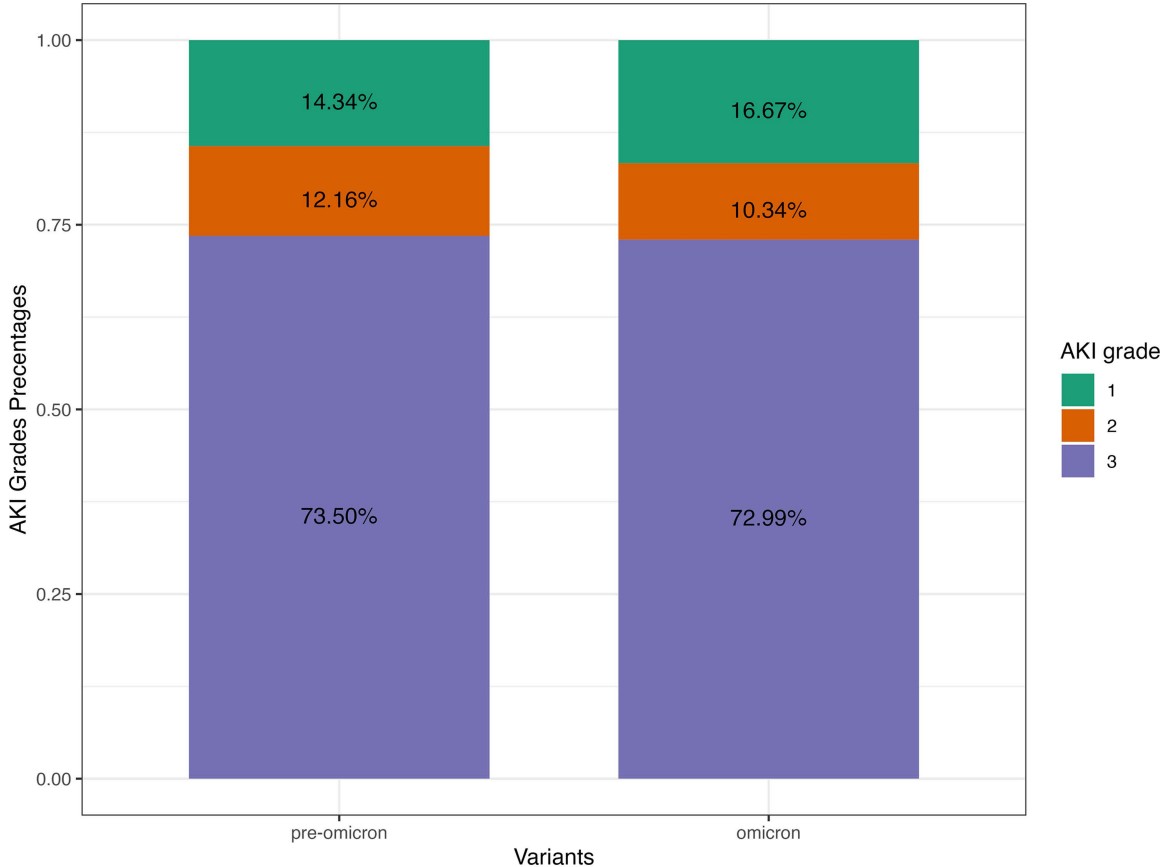

**Fig 5. AKI Grades by variants, AKI are graded according to KDIGO guideline.** AKI: acute kidney injury; KDIGO: Kidney Disease Improving Global Outcome.

in COVID-19 patients admitted to hospitals during the Delta variant period. Their recent study evaluated the same predictive model for HC-AKI using Omicron variant patients' data [35]. This study finds that the predictive model controlling for demographics, comorbidities, lab values and routine care can accurately predict the development of HC-AKI across several dominant variants strains [35]. This indicates that the etiology and risk factors for AKI have not substantially changed across these variants for hospital-acquired AKI patients [35]. Based on these findings, our study effectively examines the relationship between the Omicron variant and AKI patients. After controlling for confounders including demographics, comorbidities, lab results and treatments, there was a lower likelihood of AKI among Omicron patients compared to pre-Omicron patients holding other variables constant.

We acknowledge there are a few limitations to this study. First, the ISARIC data collection relied on the collection of information during an evolving pandemic. Consequently, some health information was not recorded due to the limited ability and capacity of healthcare workers and institutions to gather data. The ISARIC dataset includes standard clinical information for patients who were admitted with suspected or confirmed COVID-19, it is important to note that the reasons for their hospitalization have not been consistently recorded [36, 37]. Since the symptoms were moderate for the Omicron variant, some hospitalized patients might have been admitted with non-COVID conditions but subsequently tested positive for COVID-19. Therefore, we were unable to determine whether factors other than COVID-19 may have played a more significant role in the development of AKI. With the limited capacity of data collection during the pandemic, there was no

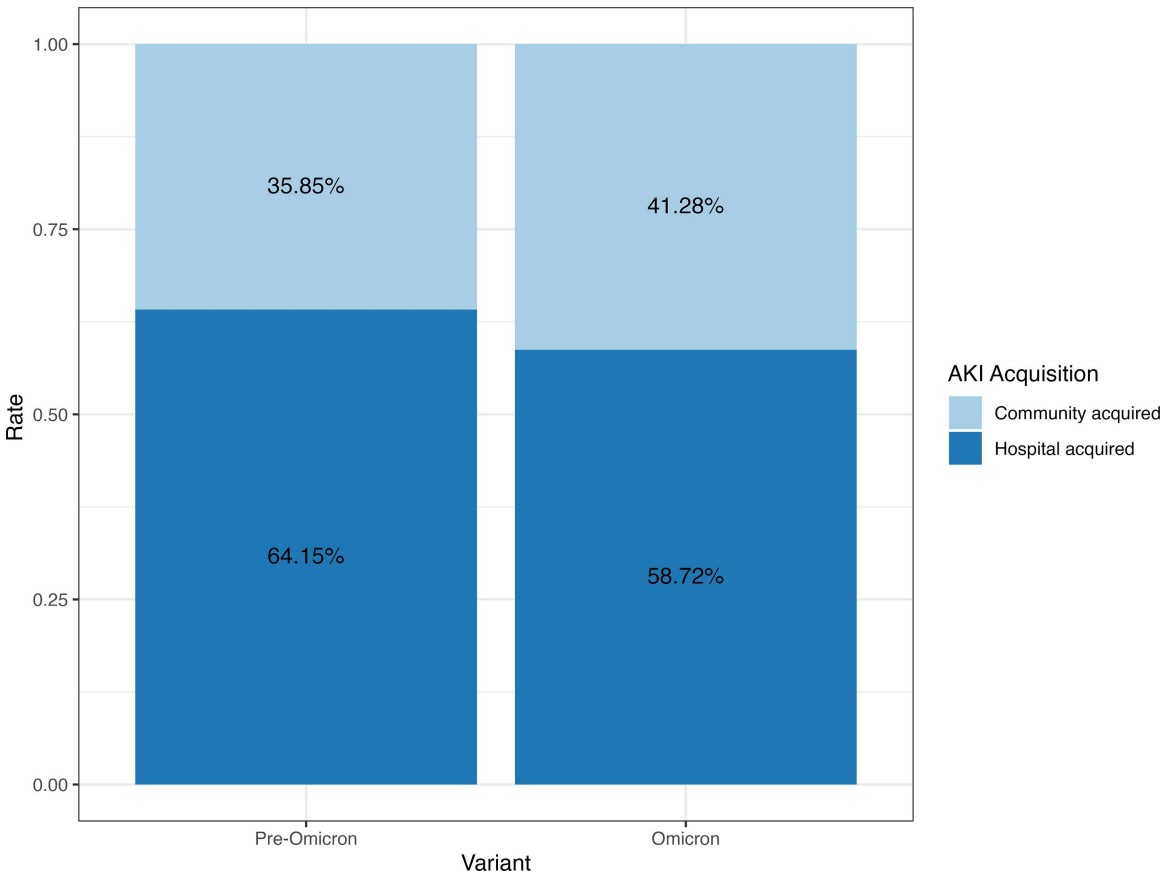

**Fig 6. Community-acquired versus hospital-acquired acute kidney injury based on 48 hour cut-off from admission by variants.** AKI, acute kidney injury.

genetic sequencing available to identify specific COVID-19 variant for each patient. Thus, we relied on the admission date where Omicron variant frequency reached 90% in the countries we included [4]. While the 90% threshold provides a strong indication, it does not eliminate the residual risk of including patients infected with variants other than Omicron. Given the ISARIC data was de-identified and no patient-level linkage across institutions, hospital transfers could not be tracked. Second, other confounders may not have been captured in this analysis including vaccination status. According to the COVID-19 vaccinations dashboard, both the UK and Spain, the two dominant countries in this analysis by patient volume, had the highest vaccine doses administered during the Omicron wave [38]. The impact of COVID-19 vaccination in preventing COVID-associated AKI is not yet clearly understood. Further study is needed to account for the vaccination status when investigating the relationship between COVID-19 variants and AKI. Third, as pre-admission creatinine levels were not available, the baseline value chosen for AKI diagnosis using the KDIGO standard was the patients' first serum creatinine within 48 hours of admission. This approach may result in an underestimation of AKI prevalence as some patients could have already experienced kidney dysfunction before hospitalisation [39]. As precise time of AKI was not recorded, we could not reliably ascertain time-to-event information for all patients. Given this analysis required multiple serum creatinine values to assess AKI, and many subjects that were missing multiple values of serum creatinine, we examined in a multivariable model the relationship between baseline characteristics comparing subjects included and those excluded due to missing multiple serum creatinine. Among baseline characteristics, there were no differences of

**Table 2. Multivariable Logistic Regression Outputs: Association between Omicron and AKI.**

| Logistic Regression | Data 1 Imputed Data | Model 1 | | | Model 2 | | | Model 3 | | |
|---|---|---|---|---|---|---|---|---|---|---|
| | *Predictors* | *Odds Ratios* | *CI* | *p* | *Odds Ratios* | *CI* | *p* | *Odds Ratios* | *CI* | *p* |
| | (Intercept) | 0.03 | 0.03–0.04 | **<0.001** | 0.04 | 0.02–0.07 | **<0.001** | 0 | 0.00–0.00 | **<0.001** |
| | Omicron | 0.66 | 0.50–0.87 | **0.004** | 0.69 | 0.50–0.94 | **0.019** | 0.78 | 0.56–1.09 | 0.142 |
| | Serum Creatinine | 1.02 | 1.02–1.02 | **<0.001** | 1.02 | 1.02–1.02 | **<0.001** | 1.02 | 1.02–1.02 | **<0.001** |
| Demographics | Age (30–39) | | | | 0.62 | 0.38–1.00 | 0.051 | 0.66 | 0.39–1.09 | 0.104 |
| | Age (40–49) | | | | 0.85 | 0.55–1.32 | 0.466 | 0.89 | 0.56–1.41 | 0.623 |
| | Age (50–59) | | | | 0.77 | 0.50–1.17 | 0.219 | 0.74 | 0.47–1.17 | 0.195 |
| | Age (60–69) | | | | 0.59 | 0.39–0.90 | **0.013** | 0.60 | 0.38–0.95 | **0.028** |
| | Age (70–79) | | | | 0.33 | 0.21–0.52 | **<0.001** | 0.39 | 0.24–0.64 | **<0.001** |
| | Age (80–89) | | | | 0.42 | 0.24–0.71 | **0.001** | 0.59 | 0.33–1.06 | 0.079 |
| | Age (>90) | | | | 0.08 | 0.02–0.37 | **0.001** | 0.08 | 0.01–0.45 | **0.004** |
| | Spain | | | | 1.04 | 0.54–2.01 | 0.903 | 2.33 | 1.13–4.81 | **0.022** |
| | The UK | | | | 0.46 | 0.25–0.85 | **0.013** | 1.05 | 0.55–1.98 | 0.892 |
| | Nepal | | | | 2.89 | 1.57–5.31 | **0.001** | 3.71 | 1.90–7.22 | **<0.001** |
| | Pakistan | | | | 2.03 | 1.13–3.65 | **0.017** | 3.73 | 1.95–7.14 | **<0.001** |
| | The US | | | | 1.31 | 0.68–2.53 | 0.414 | 1.57 | 0.80–3.09 | 0.190 |
| | Female | | | | 1.25 | 1.01–1.55 | **0.038** | 1.26 | 0.99–1.59 | 0.055 |
| Comorbidities** | Liver Disease | | | | | | | 1.06 | 0.56–2.02 | 0.857 |
| | Hypertension | | | | | | | 0.84 | 0.66–1.06 | 0.144 |
| | Chronic Pulmonary Disease | | | | | | | 1.31 | 0.85–2.02 | 0.226 |
| | Chronic Kidney Disease | | | | | | | 1.25 | 0.79–1.99 | 0.338 |
| | Chronic Cardiac Disease | | | | | | | 0.92 | 0.63–1.35 | 0.666 |
| | Diabetes | | | | | | | 0.99 | 0.69–1.42 | 0.969 |
| Lab Value | Hemoglobin | | | | | | | 1.00 | 0.99–1.00 | 0.443 |
| | Potassium | | | | | | | 1.1 | 0.93–1.30 | 0.274 |
| | Platelets | | | | | | | 1.00 | 1.00–1.00 | 0.443 |
| | WBC | | | | | | | 1.03 | 1.01–1.05 | **0.006** |
| | Sodium | | | | | | | 1.02 | 1.00–1.04 | 0.053 |
| Treatment | Mechanical Ventilation | | | | | | | 1.98 | 1.39–2.83 | **<0.001** |
| | Antibiotic | | | | | | | 2.33 | 1.42–3.81 | **0.001** |
| | Vasopressors | | | | | | | 3.30 | 2.60–4.20 | **<0.001** |
| Observations | | 3903 | | | 3903 | | | 3903 | | |
| R2 Tjur | | 0.420 | | | 0.464 | | | 0.506 | | |
| Logistic Regression | Data 2 Listwise deletion | Model 1 | | | Model 2 | | | Model 3 | | |
| | *Predictors* | *Odds Ratios* | *CI* | *p* | *Odds Ratios* | *CI* | *p* | *Odds Ratios* | *CI* | *p* |
| | (Intercept) | 0.03 | 0.03–0.04 | **<0.001** | 0.04 | 0.02–0.07 | **<0.001** | 0.00 | 0.00–0.00 | **<0.001** |
| | Omicron | 0.66 | 0.50–0.87 | **0.004** | 0.69 | 0.50–0.94 | **0.019** | 0.35 | 0.20–0.62 | **<0.001** |
| | Serum Creatinine | 1.02 | 1.02–1.02 | **<0.001** | 1.02 | 1.02–1.02 | **<0.001** | 1.02 | 1.02–1.02 | **<0.001** |

*(Continued)*

| Logistic Regression | Data 1 Imputed Data | Model 1 | | | Model 2 | | | Model 3 | | |
|---|---|---|---|---|---|---|---|---|---|---|
| | *Predictors* | *Odds Ratios* | *CI* | *p* | *Odds Ratios* | *CI* | *p* | *Odds Ratios* | *CI* | *p* |
| Demographics | Age (30–39) | | | | 0.62 | 0.38–1.00 | 0.051 | 0.90 | 0.38–2.13 | 0.818 |
| | Age (40–49) | | | | 0.85 | 0.55–1.32 | 0.466 | 1.01 | 0.46–2.19 | 0.983 |
| | Age (50–59) | | | | 0.77 | 0.50–1.17 | 0.219 | 0.77 | 0.34–1.75 | 0.53 |
| | Age (60–69) | | | | 0.59 | 0.39–0.90 | **0.013** | 0.62 | 0.28–1.34 | 0.221 |
| | Age (70–79) | | | | 0.33 | 0.21–0.52 | **<0.001** | 0.44 | 0.19–1.02 | 0.054 |
| | Age (80–89) | | | | 0.42 | 0.24–0.71 | **0.001** | 0.73 | 0.28–1.95 | 0.534 |
| | Age (>90) | | | | 0.08 | 0.02–0.37 | **0.001** | 0.15 | 0.01–1.57 | 0.113 |
| | Spain | | | | 1.04 | 0.54–2.01 | 0.903 | 0.88 | 0.27–2.84 | 0.826 |
| | The UK | | | | 0.46 | 0.25–0.85 | **0.013** | 0.78 | 0.36–1.69 | 0.531 |
| | Nepal | | | | 2.89 | 1.57–5.31 | **0.001** | 3.31 | 0.79–13.74 | 0.100 |
| | Pakistan | | | | 2.03 | 1.13–3.65 | **0.017** | 2.27 | 1.02–5.05 | **0.044** |
| | The US | | | | 1.31 | 0.68–2.53 | 0.414 | 1.26 | 0.59–2.70 | 0.555 |
| | Female | | | | 1.25 | 1.01–1.55 | **0.038** | 1.17 | 0.81–1.68 | 0.406 |
| Comorbidities** | Liver Disease | | | | | | | 1.72 | 0.74–3.98 | 0.207 |
| | Hypertension | | | | | | | 0.82 | 0.56–1.19 | 0.297 |
| | Chronic Pulmonary Disease | | | | | | | 1.20 | 0.60–2.40 | 0.608 |
| | Chronic Kidney Disease | | | | | | | 1.18 | 0.62–2.23 | 0.618 |
| | Chronic Cardiac Disease | | | | | | | 1.01 | 0.62–1.66 | 0.957 |
| | Diabetes | | | | | | | 0.99 | 0.61–1.61 | 0.966 |
| Lab Value | Hemoglobin | | | | | | | 1.00 | 0.99–1.01 | 0.835 |
| | Potassium | | | | | | | 0.95 | 0.70–1.28 | 0.723 |
| | Platelets | | | | | | | 1.00 | 1.00–1.00 | 0.635 |
| | WBC | | | | | | | 1.01 | 0.98–1.04 | 0.504 |
| | Sodium | | | | | | | 1.04 | 1.01–1.07 | **0.017** |
| Treatment | Mechanical Ventilation | | | | | | | 2.24 | 1.23–4.09 | **0.008** |
| | Antibiotic | | | | | | | 4.28 | 1.77–10.37 | **0.001** |
| | Vasopressors | | | | | | | 2.86 | 1.92–4.28 | **<0.001** |
| Observations | | 3903 | | | 3903 | | | 1863 | | |
| R2 Tjur | | 0.420 | | | 0.464 | | | 0.522 | | |
| Logistic Regression | Data 3 Imputed Data without Spain and Nepal | Model 1 | | | Model 2 | | | Model 3 | | |
| | *Predictors* | *Odds Ratios* | *CI* | *p* | *Odds Ratios* | *CI* | *p* | *Odds Ratios* | *CI* | *p* |
| | (Intercept) | 0.02 | 0.02–0.03 | **<0.001** | 0.04 | 0.02–0.09 | **<0.001** | 0.00 | 0.00–0.00 | **<0.001** |
| | Omicron | 0.58 | 0.41–0.83 | **0.003** | 0.51 | 0.35–0.76 | **0.001** | 0.56 | 0.37–0.83 | **0.005** |
| | Serum Creatinine | 1.02 | 1.02–1.02 | **<0.001** | 1.02 | 1.02–1.02 | **<0.001** | 1.02 | 1.02–1.02 | **<0.001** |

*(Continued)*

**Table 2.** (Continued)

| Logistic Regression | Data 1 Imputed Data | Model 1 | | | Model 2 | | | Model 3 | | |
|---|---|---|---|---|---|---|---|---|---|---|
| | *Predictors* | *Odds Ratios* | *CI* | *p* | *Odds Ratios* | *CI* | *p* | *Odds Ratios* | *CI* | *p* |
| Demo-graphics | Age (303132333435363738–39) | | | | 0.64 | 0.36–1.11 | 0.113 | 0.67 | 0.36–1.23 | 0.197 |
| | Age (40–49) | | | | 0.83 | 0.50–1.39 | 0.485 | 0.93 | 0.53–1.63 | 0.807 |
| | Age (50–59) | | | | 0.75 | 0.46–1.25 | 0.271 | 0.81 | 0.46–1.42 | 0.454 |
| | Age (60–69) | | | | 0.57 | 0.35–0.94 | **0.027** | 0.62 | 0.35–1.09 | 0.098 |
| | Age (70–79) | | | | 0.25 | 0.14–0.44 | **<0.001** | 0.32 | 0.17–0.60 | **<0.001** |
| | Age (80–89) | | | | 0.54 | 0.28–1.02 | 0.056 | 0.88 | 0.43–1.82 | 0.730 |
| | Age (>90) | | | | 0.12 | 0.02–0.76 | **0.025** | 0.09 | 0.01–1.06 | 0.056 |
| | Spain | | | | | | | | | |
| | The UK | | | | 0.41 | 0.22–0.76 | **0.005** | 1.01 | 0.52–1.97 | 0.983 |
| | Nepal | | | | | | | | | |
| | Pakistan | | | | 1.83 | 1.01–3.33 | **0.047** | 3.54 | 1.79–7.01 | **<0.001** |
| | The US | | | | 1.19 | 0.61–2.32 | 0.6 | 1.47 | 0.73–2.97 | 0.277 |
| | Female | | | | 1.19 | 0.92–1.54 | 0.176 | 1.25 | 0.94–1.67 | 0.123 |
| Comorbidi-ties** | Liver Disease | | | | | | | 1.24 | 0.57–2.69 | 0.581 |
| | Hypertension | | | | | | | 0.73 | 0.54–0.97 | **0.031** |
| | Chronic Pulmonary Disease | | | | | | | 1.14 | 0.64–2.03 | 0.650 |
| | Chronic Kidney Disease | | | | | | | 1.26 | 0.74–2.15 | 0.393 |
| | Chronic Cardiac Disease | | | | | | | 1.00 | 0.66–1.53 | 0.998 |
| | Diabetes | | | | | | | 1.03 | 0.70–1.52 | 0.873 |
| Lab Value | Hemoglobin | | | | | | | 1.00 | 0.99–1.01 | 0.978 |
| | Potassium | | | | | | | 1.06 | 0.85–1.31 | 0.608 |
| | Platelets | | | | | | | 1.00 | 1.00–1.00 | 0.462 |
| | WBC | | | | | | | 1.02 | 1.00–1.05 | 0.053 |
| | Sodium | | | | | | | 1.05 | 1.02–1.07 | **<0.001** |
| Treatment | Mechanical Ventilation | | | | | | | 2.53 | 1.58–4.06 | **<0.001** |
| | Antibiotic | | | | | | | 3.24 | 1.79–5.83 | **<0.001** |
| | Vasopressors | | | | | | | 3.53 | 2.62–4.76 | **<0.001** |
| Observations | | 2962 | | | 2962 | | | 2962 | | |
| R2 Tjur | | 0.434 | | | 0.475 | | | 0.526 | | |

WBC: white blood count

Baseline definition: Age (18–29), country – Canada

**: Comprehensive definitions for comorbidities, complication and outcomes can be found in S3 Table in S1 File

note apart from subjects (1) from the omicron period and (2) discharged from hospital which decreased the likelihood of inclusion (see S4 Table in S1 File). We estimate these differences would have little potential to bias the current analysis given these relationships were expected.

This study includes six countries that satisfied the selection criteria to provide a statistically reasonable sample size from pre-Omicron and Omicron variants and controls for socioeconomic factors. This implies that generalisability needs to be done with caution and might be an issue when interpreting the results.

To our understanding, this study represents the initial attempt to thoroughly investigate the relationship between the Omicron variant and the consequences of individuals experiencing AKI and COVID-19 in the ICU setting. To the best of

our knowledge, our research population constitutes the most extensive multinational group of COVID-19 ICU patients. Omicron patients admitted to ICU were older and had more comorbidities compared to pre-Omicron patients. Mortality was lower among Omicron ICU patients compared with pre-Omicron ICU patients. While the prevalence of AKI was higher in Omicron patients, when controlling for initial serum creatinine, demographics, comorbidities, lab results and treatments, the logistics regression suggests that patients admitted to the hospital with the Omicron variant had a lower risk of developing AKI than the pre-Omicron variant. As the sensitivity analysis between imputed data and listwise deletion showed different odds ratio for Omicron variant, the exact impact of Omicron on kidney function is not certain. The distribution of AKI grades was similar between pre-Omicron and Omicron surge. The relatively unnoticed occurrence of AKI in the Omicron variant brings forth adverse consequences, emphasizing the critical necessity of vigilant monitoring and prompt interventions. With the latest discovery of Omicron EG.5 [40], the challenge of understanding COVID-19 and AKI continues. Further studies are needed to better comprehend the viruses that have changed our lives in many ways and prepare us for future variants and waves of COVID-19.

## Supporting information

**S1 Data. Omicron impact on AKI Collaborators Affiliations.**
(XLSX)

**S1 File. Supplementary file for tables and figures.**
(DOCX)

## Acknowledgments

The investigators acknowledge the philanthropic support of the donors to the University of Oxford's COVID-19 Research Response Fund, and Preparedness work conducted by the Short Period Incidence Study of Severe Acute Respiratory Infection.

This work uses data provided by patients and collected by the NHS as part of their care and support #DataSavesLives. The data used for this research were obtained from ISARIC4C. We are extremely grateful to the 2648 frontline NHS clinical and research staff and volunteer medical students who collected these data in challenging circumstances; and the generosity of the patients and their families for their individual contributions in these difficult times. The COVID-19 Clinical Information Network (CO-CIN) data was collated by ISARIC4C Investigators. We also acknowledge the support of Jeremy J Farrar and Nahoko Shindo. This work is supported by the philanthropic support of the donors to the University of Oxford's COVID-19 Research Response Fund (0009109); grants from the National Institute for Health Research (NIHR; award CO-CIN-01/DH/Department of Health/United Kingdom), the Medical Research Council (MRC; grant MC_PC_19059), and by the NIHR Health Protection Research Unit (HPRU) in Emerging and Zoonotic Infections at University of Liverpool in partnership with Public Health England (PHE), (award 200907), NIHR HPRU in Respiratory Infections at Imperial College London with PHE (award 200927), Liverpool Experimental Cancer Medicine Centre (grant C18616/A25153), NIHR Biomedical Research Centre at Imperial College London (award ISBRC-1215–20013), and NIHR Clinical Research Network providing infrastructure support; the Comprehensive Local Research Networks (CLRNs) of which PJMO is an NIHR Senior Investigator (NIHR201385); Cambridge NIHR Biomedical Research Centre (award NIHR203312); CIHR Coronavirus Rapid Research Funding Opportunity OV2170359 and was coordinated out of Sunnybrook Research Institute; grants from Instituto de Salud Carlos III, Ministerio de Ciencia, Spain; the Firland Foundation, Shoreline, Washington, USA; Australian Department of Health grant (3273191); Gender Equity Strategic Fund at University of Queensland, Artificial Intelligence for Pandemics (A14PAN) at University of Queensland, The Australian Research Council Centre of Excellence for Engineered Quantum Systems (EQUS, CE170100009), The Prince Charles Hospital Foundation, Australia.

**ISARIC Clinical Characterisation Group**

Subhash Acharya, Chika Akwani, Angela Alberti, Beatrice Alex, Syed Ali Abbas, Aneela Altaf, Patrick Archambault, Rakesh Arora, Diptesh Aryal, Muhammad Sheharyar Ashraf, Namra Asif, Mohammad Asim, AM Udara Lakshan Attanyake, Benjamin Bach, John Kenneth Baillie, Wendy S. Barclay, Joaquín Baruch, Husna Begum, José Luis Bernal Sobrino, Debby Bogaert, Aidan Burrell, Kate Calligy, Gayle Carney, François Martin Carrier, Gail Carson, Alexandros Cavayas, Shelby Cerkovnik, Meera Chand, Sung-Min Cho, Barbara Wanjiru Citarella, Sara Clohisey, Marie Connor, Graham S. Cooke, Amanda Corley, Juthaporn Cowan, Juan Luis Cruz Bermúdez, Jaime Cruz Rojo, Andrew Dagens, Peter Daley, Jo Dalton, Heidi Dalton, Ana da Silva Filipe, William Dechert, Emmanuelle Denis, Thushan de Silva, Annemarie B. Docherty, Arjen M Dondorp, Christl A. Donnelly, James Joshua Douglas, Thomas Drake, Jake Dunning, Mathilde Duplaix, Lucian Durham III, Cameron J. Fairfield, Komal Fareed, Tom Fletcher, Brigid Flynn, Simon Forsyth, Robert A. Fowler, Diego Franch-Llasat, Christophe Fraser, John F. Fraser, Carrol Gamble, Julia Garcia-Diaz, Noelia García Barrio, Basanta Gauli, Praveen Kumar Ghisulal, Michelle Girvan, Margarite Grable, Christopher A. Green, William Greenhalf, Fiona Griffiths, Fakhir Raza Haidri, Matthew Hall, Sophie Halpin, Rashan Haniffa, Hayley Hardwick, Ewen M. Harrison, Janet Harrison, Alan Hartman, Madiha Hashmi, Muhammad Hayat, Ross Hendry, Rupert Higgins, Samuel Hinton, Antonia Ho, Peter Horby, Catherine L. Hough, Abby Hurd, Samreen Ijaz, Rana Imran Sikander, Clare Jackson, Denise Jaworsky, Darakhshan Kanwal, Christiana Kartsonaki, Seán Keating, Sadie Kelly, Kalynn Kennon, Imrana Khalid, Quratul Ain Khan, Sushil Khanal, Michelle E. Kho, Saye Khoo, Muhammad Nasir Khoso, Paul Klenerman, Stephen R Knight, Chamira Kodippily, Sabin Koirala, Ashok Kumar, Demetrios Kutsogiannis, Antonio Lalueza, Andy Law, Teresa Lawrence, Jennifer Lee, Gary Leeming, Gianluigi Li Bassi, Wei Shen Lim, Carlos Lumbreras Bermejo, Miles Lunn, Guillermo Maestro de la Calle, Frank Manetta, Laura Marsh, John Marshall, Sobia Masood, Mohd Basri Mat Nor, Colin McArthur, Anne McCarthy, Sarah E. McDonald, Kenneth A. McLean, Alexander J. Mentzer, Laura Merson, Dan Meyer, Alison M. Meynert, Efstathia Mihelis, Sarah Moore, Shona C. Moore, Syed Muneeb Ali, Laveena Munshi, Srinivas Murthy, Himasha Muvindi, Alex Nagrebetsky, Mangala Narasimhan, Rashid Nasim Khan, Matthew Nelder, Emily Neumann, Alistair D Nichol, Mahdad Noursadeghi, Piero L. Olliaro, Wilna Oosthuyzen, Peter Openshaw, Saijad Orakzai, Massimo Palmarini, Carlo Palmieri, Hem Paneru, Rachael Parke, Lisa Patterson, William A. Paxton, Miguel Pedrera Jiménez, Ithan D. Peltan, Riinu Pius, Georgios Pollakis, Mark G. Pritchard, Gamage Dona Dilanthi Priyadarshani, Víctor Quirós González, Aldo Rafael, Muhammad Asim Rana, Thalha Rashan, Aasiyah Rashan, Ghulam Rasheed, Ali Raza, Brenda Reeve, Atta Ur Rehman, Hongru Ren, Oleksa Rewa, Stephanie Roberts, David L. Robertson, Ferran Roche-Campo, Amanda Rojek, Clark D. Russell, Valla Sahraei, Nawal Salahuddin, Emely Sanchez, Vanessa Sancho-Shimizu, Gyan Sandhu, Marlene Santos, Shirley Sarfo-Mensah, Iam Claire E. Sarmiento, Egle Saviciute, Justin Schaffer, Gary Schwartz, Janet T. Scott, Malcolm G. Semple, Pablo Serrano Balazote, Victoria Shaw, Catherine A. Shaw, Mohiuddin Shiekh, Sally Shrapnel, Shubha Kalyan Shrestha, Louise Sigfrid, Sue Smith, Tom Solomon, B. P. Sanka Ruwan Sri Darshana, Shiranee Sriskandan, David Stuart, Jacky Y. Suen, Charlotte Summers, Hubert Tessier-Grenier, Anand Thakur, Shaun Thompson, Emma C. Thomson, Mathew Thorpe, Ryan S. Thwaites, Lance C.W. Turtle, PG Ishara Udayanga, Timothy M. Uyeki, Michael Varrone, Steve Webb, Jia Wei, Murray Wham, Nicole White, Bailey Williams, Virginie Williams, Maria Zambon.

## Author contributions

**Conceptualization:** Danyang Dai.

**Formal analysis:** Danyang Dai.

**Investigation:** Danyang Dai.

**Methodology:** Danyang Dai.

**Project administration:** Danyang Dai.

**Resources:** Danyang Dai, Marina Wainstein, Sally Shrapnel.

**Software:** Danyang Dai.

**Supervision:** Pedro Franca Gois, Sally Shrapnel, Jason D Pole.

**Validation:** Danyang Dai, Pedro Franca Gois, Marina Wainstein, Sally Shrapnel, Jason D Pole.

**Visualization:** Danyang Dai.

**Writing – original draft:** Danyang Dai.

**Writing – review & editing:** Danyang Dai, Pedro Franca Gois, Marina Wainstein, Moji Ghadimi, Nicholas Spyrison, Rolando Claure-Del Granado, Sally Shrapnel, Jason D Pole.

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
