## [Decision Letter · Decision Letter 0]

3 Aug 2025

Dear Dr. Pole,

Thank you for submitting your manuscript to PLOS ONE. After careful consideration, we feel that it has merit but does not fully meet PLOS ONE’s publication criteria as it currently stands. Therefore, we invite you to submit a revised version of the manuscript that addresses the points raised during the review process.

We appreciate the interesting study. However, there are some points raised by the reviewers. Please carefully respond to the reviewer comments and suggestions.

We look forward to receiving your revised manuscript.

Kind regards,

Vipa Thanachartwet, M.D.

Academic Editor

PLOS ONE

Journal Requirements:

“DD is supported by the University of Queensland’s Research and Training Scholarship and Digital Health UQ-QH Alliance Scholarship. SS is supported by ARC CoE Engineered Quantum Systems. DD, SS and NS declared funding from Artificial Intelligence for Pandemics (A14PAN) at University of Queensland. This work was made possible with the support of the UK Foreign, Commonwealth and Development Office and Wellcome [215091/Z/18/Z, 222410/Z/21/Z, 225288/Z/22/Z and 220757/Z/20/Z]; the Bill & Melinda Gates Foundation [OPP1209135].”

“DD is supported by the University of Queensland’s Research and Training Scholarship and Digital Health UQ-QH Alliance Scholarship. SS is supported by ARC CoE Engineered Quantum Systems. DD, SS and NS declared funding from Artificial Intelligence for Pandemics (A14PAN) at University of Queensland. This work was made possible with the support of the UK Foreign, Commonwealth and Development Office and Wellcome [215091/Z/18/Z, 222410/Z/21/Z, 225288/Z/22/Z and 220757/Z/20/Z]; the Bill & Melinda Gates Foundation [OPP1209135].”

“CDG is a member at large of the ISNB Executive Committee, co-chair of the SLANH

AKI Committee and Deputy Chair of the ISN Fellowship committee.”

5. We note that you have indicated that there are restrictions to data sharing for this study. For studies involving human research participant data or other sensitive data, we encourage authors to share de-identified or anonymized data. However, when data cannot be publicly shared for ethical reasons, we allow authors to make their data sets available upon request. For information on unacceptable data access restrictions, please see http://journals.plos.org/plosone/s/data-availability#loc-unacceptable-data-access-restrictions.

6. One of the noted authors is a group or consortium [ISARIC Characterization Group]. In addition to naming the author group, please list the individual authors and affiliations within this group in the acknowledgments section of your manuscript. Please also indicate clearly a lead author for this group along with a contact email address.

7. We note that you have included the phrase “data not shown” in your manuscript. Unfortunately, this does not meet our data sharing requirements. PLOS does not permit references to inaccessible data. We require that authors provide all relevant data within the paper, Supporting Information files, or in an acceptable, public repository. Please add a citation to support this phrase or upload the data that corresponds with these findings to a stable repository (such as Figshare or Dryad) and provide and URLs, DOIs, or accession numbers that may be used to access these data. Or, if the data are not a core part of the research being presented in your study, we ask that you remove the phrase that refers to these data.

8.Your ethics statement should only appear in the Methods section of your manuscript. If your ethics statement is written in any section besides the Methods, please move it to the Methods section and delete it from any other section. Please ensure that your ethics statement is included in your manuscript, as the ethics statement entered into the online submission form will not be published alongside your manuscript.

Reviewers' comments:

Reviewer's Responses to Questions

**Comments to the Author**

1. Is the manuscript technically sound, and do the data support the conclusions?

Reviewer #1: Yes

Reviewer #2: Yes

2. Has the statistical analysis been performed appropriately and rigorously?

Reviewer #1: Yes

Reviewer #2: Yes

3. Have the authors made all data underlying the findings in their manuscript fully available?

Reviewer #1: Yes

Reviewer #2: No

4. Is the manuscript presented in an intelligible fashion and written in standard English?

Reviewer #1: Yes

Reviewer #2: Yes

Reviewer #1: I would like to thank the editor and authors for the opportunity to review this very interesting manuscript on the Omicron Impact on Acute Kidney Injury in ICU Patients.

The manuscript is well written, the methods used allow a interpretable results which could be discussed.

We would like to highlights some points which could improve the present manuscript.

Introduction

Nevertheless, compared to the earlier Delta variant, it has been associated with higher serum creatinine levels on admission to hospital, denoting greater renal impact [8]

A single-centre study revealed a three-fold higher prevalence of AKI among intensive care unit (ICU) patients infected with the Omicron versus the Delta variant. However, the study was limited to 65 patients from a single centre who stayed in the ICU [14]

• The authors used two references that reported a high prevalence of AKI in patients in the Omicron group and developed their research hypothesis based on these references

We are not certain that all studies have reported this result, which seems to contradict the trend towards a reduction in the severity of COVID-19 over time since the appearance of COVID-19 corresponding to viral mutations, improved management with, for example, the use of corticosteroids, and the positive impact of vaccination. The authors could use the reference below (Nlandu, Y., Makulo, J. R., Essig, M., Sumaili, E., Lumaka, A., Engole, Y., … Nseka, N. (2023). Factors associated with acute kidney injury (AKI) and mortality in COVID-19 patients in a Sub-Saharan African intensive care unit: a single-center prospective study. Renal Failure, 45(2). https://doi.org/10.1080/0886022X.2023.2263583) where the authors reported a significantly higher prevalence of AKI-3 requiring dialysis during the first wave compared with others, in particular the 4th wave which corresponds to the omicron period. This could highlight the difference in the results of the various studies limited by their monocentric nature and the small size of their sample. This could further justify their study.

Methods

"Pre-Omicron" for admissions before December 1, 2021, and "Omicron" for admissions after January 1, 2022. These dates were chosen as they reflect the point at which the Omicron variant frequency reached 90% in the countries included in this study.

• Failure to specify the viral variant by genetic sequencing could introduce a bias in the selection of patients and therefore in the interpretation of results, particularly in the context of low-income countries where, in the absence of vaccination, two variants of concern may be found concomitantly during the same period. The 90% threshold does not exclude the residual risk of recording a greater number of patients with a variant other than omicron. The authors should emphasize this within the limits of their study.

There were no clinical flags in the datasets that identified specific variants, it was not feasible to discriminate patients infected with the Omicron variant from pre-Omicron variants in December 2021 thus these data were excluded

• This is not enough to exclude patients with other variants.

Discussion

The Chinese study reported one AKI patient for Omicron hospitalized patients and no AKI patient was recorded for Delta patients [8]

• This study was used to justify the possible high prevalence of renal failure in the context of Omicron in the “introduction” section of the manuscript and in the “discussion” section its interpretation changes towards a low prevalence of AKI. We feel that it should not be used in the introduction as it has not reported a trend towards a high prevalence of AKI.

Few studies have explored the risk factors associated with AKI development and COVID-19

• I'm not sure that this sentence is correct, given the number of studies on COVID-19 that we currently have.

Reviewer #2: The authors describe results of a paper investigating differences in factors related to AKI among hospitalized patients with COVID-19 during and before the Omicron variant surges using a large dataset from multiple countries. I have the following comments/suggestions:

Methods: What were the variables of interest that were imputed or excluded due to >20% missingness?

Methods: What was the rationale to use a logistic model instead of another method that would incorporate the time-to-event data, like Cox regression?

Methods: “The follow-up duration was calculated in days, starting from the first day of hospital admission and concluding with the earliest date of discharge, death or transfer to another hospital.”

Methods: Do we know to what extent the same hospitals are used in the pre-Omicron and Omicron periods?

Should follow-up also end if the patient develops the event (AKI)? Also, in the case of transfer to another hospital, if we are unable to continue to follow them up at the other hospital, do we have any indication of differences between patients who are or are not transferred and how that may or may not affect the results?

Table 2: This is a large table with the primary findings, so some clarification can be useful. 1) We can explicitly state the association that is described by the models in the title. 2) Sample sizes for each analyses can also be useful (e.g. can indicate the degree of missingness when comparing findings). 3) Three sets of analyses appear that are labeled “Data 1, Imputed data,” “Data 2, Listwise deletion,” and “Data 3, 6”. What is this last one? 4) The comparisons made or the reference group for each odds ratio are not clear. For some, though it’s not explicit, we can assume (e.g. female vs. male, diabetes vs no diabetes), though for others it’s not clear. For example, for the continuous variables and the countries. All these can be made explicit in the rows or for brevity, in the footnotes.

Given the large amount of covariates in Model 3, could it be overfit? Were propensity scores considered?

**Do you want your identity to be public for this peer review?** For information about this choice, including consent withdrawal, please see our Privacy Policy

Reviewer #1: No

Reviewer #2: No

---

## [Author Response · Author response to Decision Letter 1]

3 Sep 2025

Please see our responses to the Academic editor’s comments with journal requirements:

Journal Requirements:

Response:

We thank the editor for providing the references. We have reformatted our manuscript to align with the journal requirement.

“DD is supported by the University of Queensland’s Research and Training Scholarship and Digital Health UQ-QH Alliance Scholarship. SS is supported by ARC CoE Engineered Quantum Systems. DD, SS and NS declared funding from Artificial Intelligence for Pandemics (A14PAN) at University of Queensland.

Response:

Please see the justified funding statement below:

DD is supported by the University of Queensland’s Research and Training Scholarship and Digital Health UQ-QH Alliance Scholarship. SS is supported by ARC CoE Engineered Quantum Systems. DD, SS and NS declared funding from Artificial Intelligence for Pandemics (A14PAN) at University of Queensland. The ISARIC Clinical Epidemiology Platform used for this work was supported by the Wellcome Trust [303666/Z/23/Z]; UK International Development [301542-403]; and the Gates Foundation [INV-063472]. There was no additional external funding received for this study.

“DD is supported by the University of Queensland’s Research and Training Scholarship and Digital Health UQ-QH Alliance Scholarship. SS is supported by ARC CoE Engineered Quantum Systems. DD, SS and NS declared funding from Artificial Intelligence for Pandemics (A14PAN) at University of Queensland.

Response:

Please see the Financial disclosure below:

DD is supported by the University of Queensland’s Research and Training Scholarship and Digital Health UQ-QH Alliance Scholarship. SS is supported by ARC CoE Engineered Quantum Systems. DD, SS and NS declared funding from Artificial Intelligence for Pandemics (A14PAN) at University of Queensland. The ISARIC Clinical Characterisation Group is supported by the Wellcome Trust [303666/Z/23/Z]; UK International Development [301542-403]; and the Gates Foundation [INV-063472]. The funders had no role in study design, data collection and analysis, decision to publish, or preparation of the manuscript.

“CDG is a member at large of the ISNB Executive Committee, co-chair of the SLANH

AKI Committee and Deputy Chair of the ISN Fellowship committee.”

Response:

Please see the updated Competing Interests below:

CDG is a member at large of the ISNB Executive Committee, co-chair of the SLANH AKI Committee and Deputy Chair of the ISN Fellowship committee. This does not alter our adherence to PLOS ONE policies on sharing data and materials.

5. We note that you have indicated that there are restrictions to data sharing for this study. For studies involving human research participant data or other sensitive data, we encourage authors to share de-identified or anonymized data. However, when data cannot be publicly shared for ethical reasons, we allow authors to make their data sets available upon request. For information on unacceptable data access restrictions, please see http://journals.plos.org/plosone/s/data-availability#loc-unacceptable-data-access-restrictions.

Response:

We have updated our Data Availability statements, please see below:

The data that underpin this analysis are highly detailed clinical data on individuals hospitalised with COVID-19. Due to the sensitive nature of these data and the associated privacy concerns, they are available via a governed data access mechanism following review and approval by a data access committee. Data can be requested via the IDDO COVID-19 Data Sharing Platform (http://www.iddo.org/covid-19). The Data Access Application, Terms of Access and details of the Data Access Committee are available on the website. Briefly, the requirements for access are a request from a qualified researcher working with a legal entity who have a health and/or research remit; a scientifically valid reason for data access which adheres to appropriate ethical principles. The full terms are at https://www.iddo.org/document/covid-19-data-access-guidelines.

6. One of the noted authors is a group or consortium [ISARIC Characterization Group]. In addition to naming the author group, please list the individual authors and affiliations within this group in the acknowledgments section of your manuscript. Please also indicate clearly a lead author for this group along with a contact email address.

Response:

Please see members of the ISARIC Characterization Group attached. We would need to make each of the members a citable author. The attached file contains the members’ name and affiliations. Thank you.

7. We note that you have included the phrase “data not shown” in your manuscript. Unfortunately, this does not meet our data sharing requirements. PLOS does not permit references to inaccessible data. We require that authors provide all relevant data within the paper, Supporting Information files, or in an acceptable, public repository. Please add a citation to support this phrase or upload the data that corresponds with these findings to a stable repository (such as Figshare or Dryad) and provide and URLs, DOIs, or accession numbers that may be used to access these data. Or, if the data are not a core part of the research being presented in your study, we ask that you remove the phrase that refers to these data.

Response:

We thank the Academic editor for pointing this out. We have added the additional supporting information in the supplementary materials (S-Table 4) and addressed this in the manuscript.

8. Your ethics statement should only appear in the Methods section of your manuscript. If your ethics statement is written in any section besides the Methods, please move it to the Methods section and delete it from any other section. Please ensure that your ethics statement is included in your manuscript, as the ethics statement entered into the online submission form will not be published alongside your manuscript.

Response:

We have made sure that the ethics statement only appears in the Methods section.

Response:

We thank the editor for providing the details. We have included captions for the Supporting Information at the end of our manuscript.

Response:

We thank the editor for the suggestions. All suggested references were considered carefully and have been added with caution.

Please see our responses to the reviewers’ comments below:

Review 1:

Introduction

Nevertheless, compared to the earlier Delta variant, it has been associated with higher serum creatinine levels on admission to hospital, denoting greater renal impact [8]

A single-centre study revealed a three-fold higher prevalence of AKI among intensive care unit (ICU) patients infected with the Omicron versus the Delta variant. However, the study was limited to 65 patients from a single centre who stayed in the ICU [14]

• The authors used two references that reported a high prevalence of AKI in patients in the Omicron group and developed their research hypothesis based on these references

We are not certain that all studies have reported this result, which seems to contradict the trend towards a reduction in the severity of COVID-19 over time since the appearance of COVID-19 corresponding to viral mutations, improved management with, for example, the use of corticosteroids, and the positive impact of vaccination. The authors could use the reference below (Nlandu, Y., Makulo, J. R., Essig, M., Sumaili, E., Lumaka, A., Engole, Y., … Nseka, N. (2023). Factors associated with acute kidney injury (AKI) and mortality in COVID-19 patients in a Sub-Saharan African intensive care unit: a single-center prospective study. Renal Failure, 45(2). https://doi.org/10.1080/0886022X.2023.2263583) where the authors reported a significantly higher prevalence of AKI-3 requiring dialysis during the first wave compared with others, in particular the 4th wave which corresponds to the omicron period. This could highlight the difference in the results of the various studies limited by their monocentric nature and the small size of their sample. This could further justify their study.

Response:

We thank the reviewer for suggesting the literature, we have read the recommended paper and decided to include this reference. This reference was added to the introduction section.

Methods

"Pre-Omicron" for admissions before December 1, 2021, and "Omicron" for admissions after January 1, 2022. These dates were chosen as they reflect the point at which the Omicron variant frequency reached 90% in the countries included in this study.

• Failure to specify the viral variant by genetic sequencing could introduce a bias in the selection of patients and therefore in the interpretation of results, particularly in the context of low-income countries where, in the absence of vaccination, two variants of concern may be found concomitantly during the same period. The 90% threshold does not exclude the residual risk of recording a greater number of patients with a variant other than omicron. The authors should emphasize this within the limits of their study.

Response:

We thank the reviewer for pointing out this limitation. We had added this limitation in our discussion.

There were no clinical flags in the datasets that identified specific variants, it was not feasible to discriminate patients infected with the Omicron variant from pre-Omicron variants in December 2021 thus these data were excluded

• This is not enough to exclude patients with other variants.

Response:

We thank the reviewer for pointing this out. We acknowledge that our dataset does not include clinical or genomic markers to support the identification of specific variants. As such, we could not reliably differentiate infections caused by the Omicron variant from those caused by earlier variants, particularly during months identified by the WHO as periods of variant transition. To minimise potential sources of bias, we therefore excluded data from patients admitted in December 2021. This limitation has been acknowledged in the discussion part of the manuscript.

Discussion

The Chinese study reported one AKI patient for Omicron hospitalized patients and no AKI patient was recorded for Delta patients [8]

• This study was used to justify the possible high prevalence of renal failure in the context of Omicron in the “introduction” section of the manuscript and in the “discussion” section its interpretation changes towards a low prevalence of AKI. We feel that it should not be used in the introduction as it has not reported a trend towards a high prevalence of AKI.

Response:

We thank the reviewer for the suggestion. We have removed this reference in the introduction.

Few studies have explored the risk factors associated with AKI development and COVID-19

• I'm not sure that this sentence is correct, given the number of studies on COVID-19 that we currently have.

Response:

We thank the reviewer for pointing this out. We have revised the manuscript to more accurately reflect the current body of evidence.

Reviewer 2:

The authors describe results of a paper investigating differences in factors related to AKI among hospitalized patients with COVID-19 during and before the Omicron variant surges using a large dataset from multiple countries. I have the following comments/suggestions:

Methods: What were the variables of interest that were imputed or excluded due to >20% missingness?

Response:

We appreciate the reviewer's attention to this methodological detail. The list of variables imputed and the percentage of missingness can be found in Supplementary Figure 1. The list of variables imputed includes hypertension, Systolic Blood Pressure, liver disease, hemoglobin, potassium, sodium, white blood cell count, platelet count, mechanical ventilation, antibiotics, vasopressors, chronic pulmonary diseases, chronic kidney disease, chronic cardiac disease and diabetes. Variable of interests excluded due to >20% of missingness was bicarbonate. We have added this information in the results section.

Methods: What was the

---

## [Decision Letter · Decision Letter 1]

8 Oct 2025

Dear Dr. Pole,

Thank you for submitting your manuscript to PLOS ONE. After careful consideration, we feel that it has merit but does not fully meet PLOS ONE’s publication criteria as it currently stands. Therefore, we invite you to submit a revised version of the manuscript that addresses the points raised during the review process.

We appreciate your efforts for the study and the authors have made a careful revision to the manuscript. However, there are some minor points raised by the reviewer#1’s. Please carefully respond to the reviewer comments and suggestions.

We look forward to receiving your revised manuscript.

Kind regards,

Vipa Thanachartwet, M.D.

Academic Editor

PLOS ONE

Journal Requirements:

Reviewers' comments:

Reviewer's Responses to Questions

**Comments to the Author**

Reviewer #1: (No Response)

Reviewer #2: All comments have been addressed

2. Is the manuscript technically sound, and do the data support the conclusions?

Reviewer #1: Yes

Reviewer #2: Yes

3. Has the statistical analysis been performed appropriately and rigorously?

Reviewer #1: Yes

Reviewer #2: Yes

4. Have the authors made all data underlying the findings in their manuscript fully available?

Reviewer #1: Yes

Reviewer #2: Yes

5. Is the manuscript presented in an intelligible fashion and written in standard English?

Reviewer #1: Yes

Reviewer #2: Yes

Reviewer #1: Firstly, I would like to thank the editor and authors for giving me the opportunity to review the manuscript.

Below is our only comment regarding the latest version submitted.

Introduction section

A single-centre study revealed a three-fold higher prevalence of AKI among intensive care unit (ICU) patients infected with the Omicron versus the Delta variant. Another study identified significant higher prevalence of AKI in the 4th wave which corresponds to the Omicron period [13].

I would like to thank the authors for taking our suggestions into account and adding the recommended reference. However, they should amend the text to state that, in this study, the prevalence of AKI during the fourth wave of the pandemic was lower than in previous waves.

Reviewer #2: (No Response)

**Do you want your identity to be public for this peer review?** For information about this choice, including consent withdrawal, please see our Privacy Policy

Reviewer #1: No

Reviewer #2: No

---

## [Author Response · Author response to Decision Letter 2]

12 Oct 2025

Please see our responses to reviewer 1’s comment:

Review 1 Comment:

Reviewer #1:

Firstly, I would like to thank the editor and authors for giving me the opportunity to review the manuscript.

Below is our only comment regarding the latest version submitted.

Introduction section

A single-centre study revealed a three-fold higher prevalence of AKI among intensive care unit (ICU) patients infected with the Omicron versus the Delta variant. Another study identified significant higher prevalence of AKI in the 4th wave which corresponds to the Omicron period [13].

I would like to thank the authors for taking our suggestions into account and adding the recommended reference. However, they should amend the text to state that, in this study, the prevalence of AKI during the fourth wave of the pandemic was lower than in previous waves.

Response:

We thank the reviewer for pointing this out. We have amended the introduction to accurately reflect the referenced study. See the amended part below:

“Another study identified significant higher prevalence of AKI in the 4th wave which corresponds to the Omicron period compared to the 3rd wave [13]. In this study, the 4th wave AKI prevalence was lower compared to the first and second wave which corresponds to the beginning of COVID [13] .”

---

## [Decision Letter · Decision Letter 2]

2 Nov 2025

Omicron Surge Impact on Acute Kidney Injury in ICU Patients: A Study Using the ISARIC COVID-19 Database

PONE-D-25-17813R2

Dear Dr. Pole,

We’re pleased to inform you that your manuscript has been judged scientifically suitable for publication and will be formally accepted for publication once it meets all outstanding technical requirements.

Kind regards,

Vipa Thanachartwet, M.D.

Academic Editor

PLOS ONE

Additional Editor Comments (optional):

All comments have been addressed.

Reviewers' comments:

Reviewer's Responses to Questions

**Comments to the Author**

Reviewer #1: All comments have been addressed

2. Is the manuscript technically sound, and do the data support the conclusions?

Reviewer #1: Yes

3. Has the statistical analysis been performed appropriately and rigorously?

Reviewer #1: Yes

4. Have the authors made all data underlying the findings in their manuscript fully available?

Reviewer #1: Yes

5. Is the manuscript presented in an intelligible fashion and written in standard English?

Reviewer #1: Yes

Reviewer #1: I would like to thank the editor and authors for the opportunity to review this very interesting manuscript on the Omicron Impact on Acute Kidney Injury in ICU Patients.

All comments have been addressed, so the manuscript can now be accepted for publication.

**Do you want your identity to be public for this peer review?** For information about this choice, including consent withdrawal, please see our Privacy Policy

Reviewer #1: No

---

## [Editor Report · Acceptance letter]

PONE-D-25-17813R2

PLOS ONE

Dear Dr. Pole,

I'm pleased to inform you that your manuscript has been deemed suitable for publication in PLOS ONE. Congratulations! Your manuscript is now being handed over to our production team.

Kind regards,

on behalf of

Professor Vipa Thanachartwet

Academic Editor

PLOS ONE